# Enhanced Strength and Plasticity of CoCrNiAl_0.1_Si_0.1_ Medium Entropy Alloy via Deformation Twinning and Microband at Cryogenic Temperature

**DOI:** 10.3390/ma14247574

**Published:** 2021-12-09

**Authors:** Xiao-Hua Gu, Yu-Quan Meng, Hui Chang, Tian-Xiang Bai, Sheng-Guo Ma, Yong-Qiang Zhang, Wei-Dong Song, Zhi-Qiang Li

**Affiliations:** 1Center of Manufacture and Testing, AECC Commercial Aircraft Engine Co., Ltd., Shanghai 200241, China; guxiaohua_long@126.com (X.-H.G.); zhangyqnwpu@163.com (Y.-Q.Z.); 2Institute of Applied Mechanics, College of Mechanical and Vehicle Engineering, Taiyuan University of Technology, Taiyuan 030024, China; yqmengtyut@163.com (Y.-Q.M.); Cchanghui001@163.com (H.C.); btx1925613027@163.com (T.-X.B.); mashengguo@tyut.edu.cn (S.-G.M.); 3Shanxi Key Laboratory of Material Strength and Structural Impact, Taiyuan University of Technology, Taiyuan 030024, China; 4State Key Laboratory of Explosion Science and Technology, Beijing Institute of Technology, Beijing 100081, China

**Keywords:** CoCrNi-based medium-entropy alloys, mechanical properties, cryogenic temperature, deformation mechanisms, microband

## Abstract

The synthesis of lightweight yet strong-ductile materials has been an imperative challenge in alloy design. In this study, the CoCrNi-based medium-entropy alloys (MEAs) with added Al and Si were manufactured by vacuum arc melting furnace subsequently followed by cool rolling and anneal process. The mechanical responses of CoCrNiAl_0.1_Si_0.1_ MEAs under quasi-static (1 × 10^−3^ s^−1^) tensile strength showed that MEAs had an outstanding balance of yield strength, ultimate tensile strength, and elongation. The yield strength, ultimate tensile strength, and elongation were increased from 480 MPa, 900 MPa, and 58% at 298 K to 700 MPa, 1250 MPa, and 72% at 77 K, respectively. Temperature dependencies of the yield strength and strain hardening were investigated to understand the excellent mechanical performance, considering the contribution of lattice distortions, deformation twins, and microbands. Severe lattice distortions were determined to play a predominant role in the temperature-dependent yield stress. The Peierls barrier height increased with decreasing temperature, owing to thermal vibrations causing the effective width of a dislocation core to decrease. Through the thermodynamic formula, the stacking fault energies were calculated to be 14.12 mJ/m^2^ and 8.32 mJ/m^2^ at 298 K and 77 K, respectively. In conclusion, the enhanced strength and ductility at cryogenic temperature can be attributed to multiple deformation mechanisms including dislocations, extensive deformation twins, and microbands. The synergistic effect of multiple deformation mechanisms lead to the outstanding mechanical properties of the alloy at room and cryogenic temperature.

## 1. Introduction

High-entropy alloys (HEAs) and medium-entropy alloys (MEAs) are a class of multi-principle elements advanced structure materials that exhibit remarkable mechanical properties over conventional alloys [1,2,3]. These alloys tend to form simple solid solution phases, such as face-centered cubic (FCC), body-centered cubic (BCC), or hexagonal close-paced (HCP) structures since their high-mixing entropy can decrease the Gibbs free energy and retard the formation of intermetallic. Recently, the H-MEAs with single-phase FCC structure and low stacking fault energies (SFEs) have attracted wide attention from researchers due to their exceptional combination of high strength and ductility, high work-hardening rate, and fracture toughness. In particular, excellent mechanical performance at low temperature, such as the CoCrFeMnNi HEA and CoCrNi MEA, show strength as well as plasticity enhancement simultaneously. M-HEAs were demonstrated to be the potential candidates for cryogenic engineering structural applications such as in the nuclear reaction industry, aerospace industries, gas turbine engines, and space exploration [4,5,6].

However, CoCrFeMnNi HEA and CoCrNi MEA with medium grain sizes all perform the lower yield strength at ambient temperature, which limits their actual engineering applications [6,7]. It has been pointed out that the additions of alloying elements appropriately that induce lattice distortion is an effective strategy for microstructural design in single-phase alloys, which has been confirmed in some studies of Al, Ti, Mn, and C addition in H-MEAs [8,9,10,11,12,13]. Nevertheless, because of strength and ductility’s mutually conflicting relationship, the increase in strength is generally accompanied by a decrease in plasticity, which is called trade-off of strength and plasticity. Recently, simultaneous enhancement of strength and ductility of CrCoNiSi_x_ MEAs were proposed, which can be attributed to the decreases of SFE with the increase of Si content [14]. This provided a heuristic strategy to break this trade-off by mixing aluminum and silicon just as in the case of the first-generation twinning induce plasticity (TWIP) steel.

Meanwhile, intense efforts have been devoted to clarifying microstructure deformation mechanisms for the simultaneous enhancements of strength and plasticity of H-MEAs at low temperature [15,16,17,18]. For instance, Otto et al. [19] investigated the influence of temperature and microstructure upon the tensile properties of CoCrFeMnNi HEAs. Their results suggested that during the inception phase of plasticity, deformation occurs by the glide of 1/2<110> dislocations dissociated into 1/6<112> Shockley partials on normal FCC slip system {111}<110>. Subsequently, nano-scale deformation twins as an additional deformation mechanism give rise to the dynamic Hall–Petch effect, which induces a high degree of work hardening and a substantial improvement of ultimate tensile strength at 77 K. Furthermore, Laplanche et al. [6] carefully compared the mechanical properties, dislocation substructure, SFE, and critical twinning stress between MEA CrCoNi and HEA CrMnFeCoNi at 293 K and 77 K. It was demonstrated that the lower SFE favored the appearance of nanotwinning earlier, which provides high, steady work hardening, leading to the superior mechanical properties for CrCoNi MEAs. Moreover, Ding et al. [20] reported the exceptional damage tolerance of CrCoNi-based M-HEAs due to the synergy of deformation mechanisms, including twinning, partial and complete dislocation, cross-slip, and multiple dislocation-activated slips and twinning deformation of grain-boundary interactions at cryogenic temperature (93 K) conducted in in situ TEM straining experiments.

The previous studies primarily focused on the microstructure evolution of dislocation structure, stacking fault (SF), and deformation twinning to explain the superior strain hardening and significant ultimate tensile strength owing to the mechanical property and deformation mechanisms of HEAs being strongly dependent on the temperature and microstructure [6,7,13,14]. These studies provide extensive evidence that deformation twinning contributes to the improvement of strength and plasticity simultaneously when the M-HEAs deformed at 77 K and suggested that the index SFEs will drop when the temperature decreases, which is favorable for dislocation glide to the formation of twins and even inducing the martensite phase transformation. However, the temperature dependence of yield strength for current H-MEAs is rarely discussed in detail, while this is distinct from pure FCC alloys. This is because the nature of the strengthening mechanism is not completely understood and still controversial. In particular, the main contributions of yield stress include solute induced lattice friction (Peierls stress due to matrix atoms concerned with dislocation core structure) and solute strengthening (pinning due to solute atoms). Because of the atomic size misfit, modulus mismatch, and chemical complexity of multicomponent alloys, there is a blurring of the boundary between matrix and solute atoms. Furthermore, the analysis of the nature of barriers to dislocation motion was investigated by thermal activation processes of H-MEAs, with the study supporting the presence of nanoscale heterogeneity such as co-clusters and short range orders (SRO), which are associated with the strong strain rate and temperature dependences [21,22].

In this study, CoCrNiAl_0.1_Si_0.1_ MEA after cool rolling and anneal were fabricated to conduct a uniaxial tensile experiment at room and cryogenic temperatures. The basic idea of alloy design is that addition of Al can increase the strength, and Si enables SFE to decrease effectively, which in this respect, the modification of lattice friction response by appropriate additions of alloying elements is an effective strategy for microstructural design in single-phase alloys [23]. Moreover, the addition of low-density elements increases the specific strength of the alloy. The newly developed alloys have an excellent performance in increasing strength without loss of plasticity at room temperature compared with CoCrNi. The yield strength, ultimate tensile strength, and elongation were increased from 405 MPa, 790 MPa, and 55% to 480 MPa, 900 MPa, and 58%, exhibiting a pronounced strengthening [14]. This paper emphasizes the analyses of lattice distortion effect and twinning formation to understand the potential deformation mechanisms, including temperature-dependent yield stress and strain hardening. It is significant to explain the deformation behavior of CoCrNi-based MEAs containing trace elements at low temperatures, thus widening their industrial applications in extreme conditions.

## 2. Experiment and Methods

The ingots with a normal composition of CoCrNiAl_0.1_Si_0.1_ (atom %) were fabricated in a Ti getter high-purity argon atmosphere by vacuum arc-melting furnace (WuKeGuangDian, Beijing, China) melting a mixture of high-purity elements (purity > 99.9 wt %). The ingot was smelted five times to ensure chemical element homogeneity. A plate material with a dimension of 100 × 20 × 2 mm^3^ was obtained by water-cooled copper mold. The casted MEAs were homogenized at 1100 °C for 5 h. By cold rolling, the thickness of the plates was reduced by 70%, followed by annealing at 900 °C for 1 h with subsequent water quenching. The CoCrNi alloy was also prepared for comparison under the same experimental process, exhibiting an equiaxed grain structure with an average grain size of 16 mm. The dog bone-shaped tensile samples with the dimensions of 10 × 4 × 0.6 mm^3^ were processed by electric discharge machining for mechanical testing, and the tensile direction was along the rolling direction. Before the test, each surface of the samples was mechanically polished using 800-grit SiC paper to ensure parallelism.

The uniaxial tensile test was produced by GOTECH testing machines (Gotech Testing Machines, Qingdao, China) at a strain rate of 1 × 10^−3^ s^−1^ at room temperature (298 K) and cryogenic temperature (77 K). The engineering strain is the displacement of crosshead, and the true stress and true strain are calculated according the following formations:(1)$$\sigma_{T}=\sigma_{E}\left(1+\varepsilon_{E}\right)$$
(2)$$\varepsilon_{T}=\ln\left(1+\varepsilon_{E}\right)$$
where the $\sigma_{T}$ and $\varepsilon_{T}$ are the true stress and true strain, respectively, and${\;\sigma }_{E}$ and $\varepsilon_{E}$ are the engineering stress and engineering strain, respectively. During the cryogenic temperature test, the sample was immersed in liquid nitrogen for 15 min until the end of the test. Each test was repeated at least three times to ensure reliability.

Phase identification was conducted using X-ray diffraction (PHILIPS, Beijing, China) (XRD, PHILIPS APD-10D) through 2θ from 40° to 100° with a speed of 4°/min. Microstructure characteristics were studied using the JEOL JSM-7100F scanning electron microscope (SEM) (JEOL, Beijing, China) equipped with an electron backscattering diffraction (EBSD) detector. The EBSD specimens were prepared by mechanical polishing with a suspension of Al_2_O_3_ with a particle diameter of 50 nm and electrolytic polishing at room temperature in HClO_4_ (10%) and CH_3_COOH (90%) solutions. The FEI Talos F200X transmission electron microscopy(TEM)(FEI, Beijing, China) were used to carry out the more refined microstructures analysis at an acceleration voltage of 200 kV. The TEM sample was first mechanically ground to a thickness of 50 μm and then thinned using double-jet electropolishing in a solution of 95% ethanol and 5% perchloric acid at −20 °C and an applied voltage of 30 V, followed by Ar ion milling. All the TEM specimens after deformation were cut from the uniform deformation regions close to the fracture surface.

## 3. Results and Discussion

### 3.1. Initial Microstructure Analyses

The EBSD inverse pole figure (IPF) map in Figure 1a and the twin band contrast (BC) map in Figure 1b of MEA showed that the alloy is a completely recrystallized microstructure with random orientation and multiple micron annealing twins (as shown in the red lines). This is due to the treatment of annealing that causes a decrease in interfacial energy from interfacial migration. Moreover, the abundance of annealing twins suggests that the MEA had low SFE. Figure 1c shows the grain size distribution of the microstructure with an average grain size of 20 μm. Considering that the twin boundaries and the high angle boundaries played the same role in the plastic deformation of the alloy, they should be treated to determine the grain size equally. Analysis of the XRD (X-ray diffraction) in Figure 1d shows the phase structure of alloy was composed of simple FCC solid solution with the lattice constant at 3.573$\;\dot{A}$, a mean serious lattice distortion compared with CoCrNi (3.567$\;\dot{A}$). In addition, EDS analysis of CoCrNiAl_0.1_Si_0.1_ is shown in Table 1 to indicate the atom percentage.

### 3.2. Mechanical Properties

Figure 2a,b shows the engineering and true tensile stress–strain curves of CoCrNiAl_0.1_Si_0.1_ at room temperature (298 K) and cryogenic temperature (77 K). The yield strengths were determined to be about 480 MPa and 700 MPa, respectively, while the ultimate tensile strengths were determined to be about 950 MPa and 1250 MPa, respectively. Meanwhile, the elongation reached 58% and 72% for the alloys tested at 298 K and 77 K, respectively. Table 2 provides intuitive data. Although the simultaneous enhancement of strength and ductility of CoCrNi has been reported, it is inspirational that CoCrNiAl_0.1_Si_0.1_ has an excellent combination of strength and ductility compared with CoCrNi, CoCrNiFe, CoCrNiFeMn, FeCoCrNiAl_x_ [6,8,9,10], high-Mn steels [11], austenitic stainless steels [24], and high-Ni steels [25], particularly at 77 K, as shown in Figure 3.

The strain hardening rate (SHR) is defined as dσ/dε, which is the derivative of true stress for true strain as the function of true strain and true stress, as shown in Figure 2c,d. Upon cryogenic temperature loading, the MEA exhibited a much stronger strain hardening ability than at room temperature loading. At 298 K, the SHR showed a monotonic decay with increasing strain. In contrast, the stable strain hardening stage appeared obviously at 77 K, and the high ductility was ascribed to the high work hardening capability. Among the extensively investigated mechanical behaviors of the FCC H-MEAs, it is noted that there are three distinguishable stages of work hardening at cryogenic temperature [12]. The first stage was characterized by a continuous decrease in the strain hardening rate at ≈10% true strain, similar to that observed at 298 K. At larger strains (10%–50%), a second stage appeared, wherein the strain hardening rate remained almost constant at around 3.0 GPa. Finally, in the third stage, the strain hardening rate again decreased until failure at ≈52%. As shown in Figure 2d, the grey area indicates the region in which necking was predicted to occur according to Considere’s criterion: dσ/dε < σ. At 77 K, the SHR was consistently higher than at 298 K and its intersection with the necking line occurred at higher strain. It is noted that when stretched at 298 K, the strain value of necking onset was 41%, and at 77 K, it reached 52%. This is a consequence of the postponement of necking, which is consistent with the observation in the NiCoCr MEA upon quasi-static tension at liquid nitrogen temperature (77 K) and room temperature (293 K) [6].

### 3.3. Microstructure Characterization

Transmission electron microscopy (TEM) was applied to characterize the microstructures of CoCrNiAl_0.1_Si_0.1_ MEAs under uniaxial tension to fracture. Figure 4a,b,d,e shows the bright field (BF) TEM under room temperature and cryogenic temperature, respectively. The corresponding selected area electron diffraction (SAED) pattern along Z = [110] is displayed in Figure 4c,f. The post-deformation microstructure at room temperature is shown in Figure 4a, where dislocation cells (DCs) and high-density dislocation walls (HDDWs) are marked by red arrows, indicating the transition of dislocation structures from planar slip to wavy slip. A similar phenomenon was reported in some TWIP steels and some H-MEAs. This was because when the initial slip surfaces are obstructed, a new slip plane must be activated to accommodate the strain misfit so that dislocations are entangled at multiple slip surfaces [26]. Therefore, the grains were divided into smaller cells, leading to the relaxation of local strain concentration and homogenization of the plastic deformations that cause the strain hardening within the alloys [27]. In Figure 4b, a small number of deformation twins were observed, as indicated by yellow arrows. Twins thickness and spacing were both relatively wide. Dislocations were found to pile up against twin boundaries, which hindered dislocation transmission. Figure 6c is the corresponding selected area electron diffraction (SAED) pattern that confirms the twin structure.

In contrast to room temperature deformation shown in Figure 4d, extensive deformation twins were found to interact with dislocations effectively. The interaction accumulated dislocations within twin lamellae and generated high local stresses at twin tips that exceeded the critical stress of twin nucleation, thereby resulting in more twins [28]. As shown in Figure 4e, high density of deformation twin lamellae was observed along with a primary twinning system upon cryogenic temperature. It is indicated that there was a denser and finer distribution of deformation twins with a decrease in temperature. The twin–twin interactions can also be readily observed; secondary twins are marked by green arrows, being able to form a complex three-dimensional network restrain dislocation propagation [29]. The twin boundaries were observed to impede dislocation glide on the slip planes. Such obstructs can produce strengthening and increase the strain hardening capability of alloys [30]. Multiple twinning tended to form in some alloys with low SFE, which not only obstructed the dislocation motion to enhance the strength and strain hardening capacity of the alloy, but also facilitated multiple slips in dislocation to the accommodation of larger plastic deformation. Figure 4f is the corresponding SAED pattern to prove the twin structure.

Figure 5 shows the high-resolution transmission electron microscopy (HRTEM) images revealing the microstructure in the detail of atomic level for the deformation specimen at cryogenic temperature. In Figure 5a, nano-twinning (marked by white arrows) and numerous SF (indicated by green arrows) can be observed. In Figure 5b, the twinning structure, which was proved by IFFT (inverse Fourier transform) and twinning boundary (TB), can be distinguished visibly, and the TB was constituted by some SFs. Widespread SFs demonstrated that the MEAs had a much higher propensity to form SFs at low temperature; however, the reinforcement and strain hardening impacts of SFs may not be as remarkable as TBs. According to previous studies, the high density of SFs itself can also lead to the strengthening and strain hardening in which SFs are often regarded as obstacles to the movement of dislocations and can accumulate dislocations around them [30].

Moreover, microbands were found in the sample fractured at 77 K; as shown in Figure 6, the SEAD along <110> zone axis revealed no twinning spots, ruling out the possibility that these structures are twins. These structures consisted of bands on the order of 10 to 100 nm in width aligned along {111} planes in FCC metal, which has been widely observed in aluminum alloys, lightweight austenitic steels, duplex steels, and HEAs, especially under dynamic loading and low-temperature conditions [12,31]. Some studies suggested that the formation of microband is related to high SFE materials such as nickel and aluminum. Recently, Wang et al. [12] reported that the microbands were founded in the 23 mm grain-sized carbon-doped Fe_40.4_Ni_11.3_Mn_34.8_Al_7.5_Cr_6_ HEA. It seems reasonable to expect microbands to appear in the current MEA, which had added aluminum and silicon. Furthermore, Wang et al. [31] reported the formation of microbands but not twins in carbon-doped Fe_40.4_Ni_11.3_Mn_34.8_Al_7.5_Cr_6_ HEA when compressed at temperatures ranging from 77 to 673 K. Foley et al. [32] investigated CoCrFeMnNi under quasistatic and dynamic compression, finding that deformation at 8 × 10^3^ s^−1^ led to the formation of both twins and microbands. These studies indicate that microbanding occurred as a result of either a decreased SFE or an increase in the Peierls barrier, both consequences of decreased temperature. This clearly explains the current observations of the formation of microbands [32].

### 3.4. The Temperature Dependence Yield Strengths

The yield stress of BCC metal is strongly dependent on temperature because the Peierls–Nabarro stress is the main barrier to the movement of dislocations at low temperature. By contrast, the yield strengths of pure FCC metals are relatively insensitive to changes in temperature. The delivery of dislocation and forest dislocation is the main obstacle to dislocation movement. Heat rarely helps dislocations overcome these obstacles effectively. However, a series of studies have revealed that equiatomic alloys also exhibit varying degrees of strengthening at lower temperature—for example, NiCoCr, FeNiCoCr, and FeNiCoCrMn exhibited significant strengthening as the temperature was decreased into the cryogenic range [7,9]. The alloys studied in the present paper had the same temperature dependence, wherein the yield stress increased from 480 to 700 MPa with the temperature decrease from 298 to 77 K. This abnormal temperature dependence is markedly different from that of pure FCC metals, but closed to binary FCC alloys, which implies that the existence of stronger obstacles to dislocation motion in the distorted lattice of H-MEAs.

Despite the apparent importance of solid solutions for the strengthening of metallic alloys, it is not so obvious as to how to explain the physical mechanisms behind these phenomena in the case of H-MEAs [16,33]. Solid solution strengthening in metallic alloys manifests due to either direct or indirect interactions between solute atoms and dislocations. Due to there being no obvious difference between solute and solvent in MEA/HEA and equiatomic alloys, the local stress field created by severe lattice distortion in HEAs could strongly hinder the movement of the dislocations. As a result, it is difficult to distinguish the solute strengthening (pinning due to solute atoms or dislocation line) and the lattice friction (Peierls barriers) contributions to the yield strength [34]. Nevertheless, some theories based on lattice distortion models that originate from the elastic mismatch and the atomic size mismatch have arisen to describe the strengthening mechanism. The simplest strategy is that Wu et al. [7] assumed the complex lattice structure to be an average solvent or effective medium, speculating that the strong temperature sensitivity should be attributed to the strong lattice friction that represents average resistance offered by all the constituent atoms.

In general, the yield stress is a combination of the frictional stress ($\sigma_{fr}$), or the intrinsic lattice resistance to dislocation motion, plus the various incremental strengthening contributions, such as those due to the initial dislocation density ($\sigma_{pi}$), solid solution hardening ($\sigma_{ss}$), and grain boundary (Hall–Petch) strengthening ($\sigma_{gb}$). A general expression for the yield strength can be written as:(3)$$\sigma_{y}=\sigma_{fr}+\sigma \;{\;}_{pi}+\sigma_{ss}+\sigma_{gb}$$

Considering that there is no obvious difference between solute and solvent in MEA/HEA equiatomic alloys, the reinforcement term caused by solid dissolved atoms can be incorporated into lattice friction term [9]. In addition, the contribution of dislocation could be neglected, owing to the low initial dislocation density. Therefore, the equation is as follows:(4)$$\sigma_{y}\left(T\right)=\sigma_{fr}\left(T\right)+\sigma_{gb}\left(T\right)$$

According to the report, grain boundary strengthening did not significantly contribute to the temperature dependence of the observed yield strength. Grain boundary strengthening is controlled by the Hall–Petch effect, and the Hall–Petch relation is as follows:

$\sigma_{gb}=kd^{-1/2}$. *k* is the Hall–Petch coefficient that adopts 265 MPa·μm^−1/2^ reference CoCrNi, and *d* is the grain size, that is 20 μm. Then, the contributions are calculated, $\;\sigma_{gb}=59.2\mathrm{MPa}$, $\sigma_{fr}\left(298K\right)=420.7\mathrm{MPa}$, $\sigma_{fr}\left(77K\right)=640.7\mathrm{MPa}$.

For FCC H-MEAs, the strong Peierls stresses originate from varying potential barriers of dislocations during movement through distorted lattices, while in BCC metals, the strong Peierls stresses arise from their own special dislocation core structures (non-planar dislocation cores) [14]. Lattice distortions (*δ*) for the alloys are calculated as [25]:(5)$$\delta =\sqrt{\sum_{i=1}^{n}x_{i}{\left(1-\frac{r_{i}}{\bar{r}}\right)}^{2}}$$
where $x_{i}$ is atomic fraction, and $r_{i}$ is the Goldschmidt radius of the ith element; the value of the average atomic radius was estimated using Vegard’s law given by the formula $\bar{r}=\overset{n}{\underset{i=1}{\sum }}x_{i}r_{i}$. The lattice distortion exhibited large values for CoCrNiAl_0.1_Si_0.1_ compared with CoCrNi, as shown in Figure 7. It is indicated that adding large atomic radius elements such as aluminum will provide higher atomic misfits, causing significant distortion of the matrix.

It is generally believed that due to the difference in the lattice structure, pure BCC metals have a narrower dislocation width compared with FCC metals, which leads to a higher Peierls barrier and stronger temperature dependence of strength. For the equiatomic alloys, the width of dislocation is between them, and thus significant temperature dependence of strength is exhibited. The intrinsic barrier that the dislocation has to overcome to the dislocation motion is the lattice friction stress, also known as Peierls–Nabarro stress (P-N), which is given by Wu et al. [7]:(6)$$\sigma_{p}=\frac{2G}{1-\upsilon }\exp[\frac{-2\pi \omega_{0}}{b}\left(1+\alpha T\right)Z]$$
where *G* is the shear modulus, *ν* is Poisson’s ratio, $\omega_{0}$ is the dislocation width at 0 K, *α* is a small positive constant, *T* is the absolute temperature, and *b* is the magnitude of the Burgers vector. It can be pointed out that the dislocation width ω will fluctuate with the change of temperature, which means that dislocation width ω is associated with thermally activated processes that control dislocation mobility, and thermal vibrations cause the effective width of a dislocation core to increase. As a result, the yield stress of current alloys increase with the temperature decrease to 77 K [7].

Moreover, the study base on thermally activated deformation analysis indicated that the presence of short-range co-clusters and/or short-range orders that, owing to the enthalpy-driven atomic rearrangements during thermomechanical processing, may be the nature of thermal obstacles in terms of dislocation motion, despite there being rare direct imaging evidence of co-clusters and SROs in the current image analysis technology. A problematic concern is the existence or lack of nanoscale co-clustering and short-range ordering in HEAs. Several studies have been conducted to investigate the influence of these short-range orders based upon density functional theory calculations and large-scale molecular dynamics [33]. In complex multi-component solid solutions alloys, co-clustering and/or SRO should be considered to be that which could be attributed to the temperature dependence of yield strength [21].

### 3.5. The Temperature Dependence of Strain Hardening

In investigating the temperature dependence of strain hardening, the work-hardening part of the true stress–strain curve is defined as the value of the total stress (*σ*) minus the yield stress ($\sigma_{y}$). Strain-hardening behavior of M-HEAs is described by classical forest dislocation hardening, where the flow stress is proportional to the square root of the dislocation density [34]:(7)$$\sigma_{w}=M\alpha Gb\sqrt{\rho }$$
where *M* is the average Taylor factor, α is the numerical constant related to dislocation interaction strength, *G* is the shear modulus, *b* is the magnitude of Burgers vector for perfect dislocation, and *ρ* is the total dislocation density. Numerous studies have shown that with the increase of strain, there is a high density of dislocation accumulations as an approach to improve the strength and ductility simultaneously [35].

It is acknowledged that nano-twins provide significant strain hardening in some low-SFE alloys, which can provide barriers to dislocation glide, thereby increasing dislocation storage and decreasing their mean free path [21,36]. The steady nucleation of newly deformation twins leads the grains to fracture in the form of twin lamellae with increasing strain. As the deformation twins gradually divide the grains into smaller microstructural entities, the average free path of dislocation is reduced, as observed in Figure 4b,e. In the case of low temperature, besides interaction between twins and dislocations, more secondary twins and SF were found. The formation of the primary and secondary nano-twins effectively sub-divided the initial coarse grains into smaller sub-grains, which caused the dynamic Hall–Petch effect. It has been reported that the latent hardening from twin–twin interaction is at least one order of magnitude higher than other interactions related to slip [14,37]. Hence, twins simultaneously contribute to the high strength and strain hardening in CrCoNiAl_0.1_Si_0.1_ MEA at cryogenic temperature.

The activation of twinning is associated with SFE, which is calculated as follows:(8)$$\gamma_{\mathrm{SFE}}=2\rho {\Delta G}^{\gamma \to \varepsilon }+2\sigma$$
where *σ* is the interfacial energy per unit area of the phase boundaries, and *ρ* is the molar surface density along {111} planes, calculated by Allain et al. as follows [8]:(9)$$\rho =\frac{4}{\sqrt{3}}\frac{1}{a^{2}N}$$
where *a* is the lattice parameter of the alloy and *N* is Avogadro’s number; ${\Delta G}^{\gamma \to \varepsilon }\;$is the molar Gibbs energy of the *γ*-austenite to ε-martensite phase transformation, which can be express as
(10)$$\Delta G^{\gamma \to \varepsilon }=\underset{i}{\sum }X_{i}\Delta G_{i}^{\gamma \to \varepsilon }+\frac{1}{2}\underset{ij}{\sum }X_{i}X_{j}\Omega_{ij}^{\gamma \to \varepsilon }.\;\;\;\;\;\;\;\;\;\;\;\;\left(i,j=Co,Cr,Ni,\;Al,\;Si;i\neq j\right)$$
where *X_i_* and ${\Delta G}_{i}^{\gamma \to \varepsilon }$ represent the molar fraction and the difference of free energy between FCC and HCP phase transformation of the pure element, respectively, and $\Omega_{ij}^{\gamma \to \varepsilon }$ is an interaction energy parameter for components *i* and *j*. The relevant thermodynamic parameters are given in Table 3.

The γ_SFE_ of CoCrNiAl_0.1_Si_0.1_ MEA were calculated to be 14.12 mJ/m^2^ and 8.32 mJ/m^2^ at 298 K and 77 K, respectively. The calculation results demonstrate that the SFE decreased with the decrease of temperature.

The previous study indicated that the microband-induced plasticity (MBIP) was similar to twinning-induced plasticity (TWIP) and transformation-induced plasticity (TRIP) [31,32]. Guo et al. investigated the deformation mechanism of Cr_10_Mn_50_Fe_20_Co_10_Ni_10_ HEA and observed the activation of microbands, indicating that microbands act as dislocation sources as well as dislocation barriers, eventually leading to the formation of dislocation cell structures [32]. According to the dislocation boundary splitting model, grains were subdivided into misorientated domains to accommodate the plastic strain. Double-wall dislocation structures were gradually formed in domain boundaries via a multi-slip process to shield the stress, which is responsible for the improvement in the work hardening rate. The refinement of the microbands spacing occurred as strain continuously proceeds, which increased the passing stress acting on microbands and thus promoted strain hardening.

Accordingly, the schematic diagrams of the deformation mechanism at 298 K and 77 K are shown in Figure 8. At 298 K, the main deformation mechanisms were the dislocation cells, deformation twins, and interactions of twins and dislocation. At 77 K, it can be attributed that multiple mechanisms to the high-density dislocation increased volume fraction of nano-twins, secondary nano-twins, and microbands, which together facilitated the plastic deformation and strain hardening, thereby leading to the outstanding mechanical properties of the present alloy at low temperature, simultaneously triggering diverse strengthening effects.

The alloys exhibited a high performance in mechanical properties at 77 K and 298 K, which made them very attractive to cryogenic temperature structural applications for achieving increased energy efficiencies in a broad of engineering fields, such as aerospace, space exploration, nuclear power, and many other industries.

## 4. Conclusions

In this study, CoCrNiAl_0.1_Si_0.1_ MEA manufactured by a vacuum arc-melting furnace was investigated. Mechanical properties of the CoCrNiAl_0.1_Si_0.1_ MEA at room temperature (298 K) and cryogenic temperature (77 K) were produced. The temperature dependence of yield stress and strain hardening behavior and corresponding microstructural features during tensile deformation were investigated. The main conclusions are as follows:CoCrNiAl_0.1_Si_0.1_ exhibited remarkable simultaneous enhancement of strength and ductility at low temperature. The yield strength increased from 480 MPa to 700 MPa; moreover, the ultimate tensile strength increased from 950 MPa to 1250 MPa. Meanwhile, the elongation reached from 58% to 72% tested at 298 K and 77 K, respectively.CoCrNiAl_0.1_Si_0.1_ severe lattice distortion inducted Peierls lattice friction stress. The Peierls barrier height increased with decreasing temperature, owing to thermal vibrations, causing the effective width of a dislocation core to decrease. Meanwhile, the presence of co-clustering and/or SRO were the sources of thermal obstacles to dislocation motion, which contributed to the temperature dependence of yield stress.At 298 K, the main deformation mechanism of CoCrNiAl_0.1_Si_0.1_ was the dislocation cells, deformation twins, and interactions of twins and dislocation. At 77 K, it was attributed to multiple mechanisms work together to the high-density dislocation, the increased volume fraction of nano-twins, secondary nano-twins, and microbands, which facilitated the continuous strain hardening and remarkable plastic deformation.

## Figures and Tables

**Figure 1 materials-14-07574-f001:**
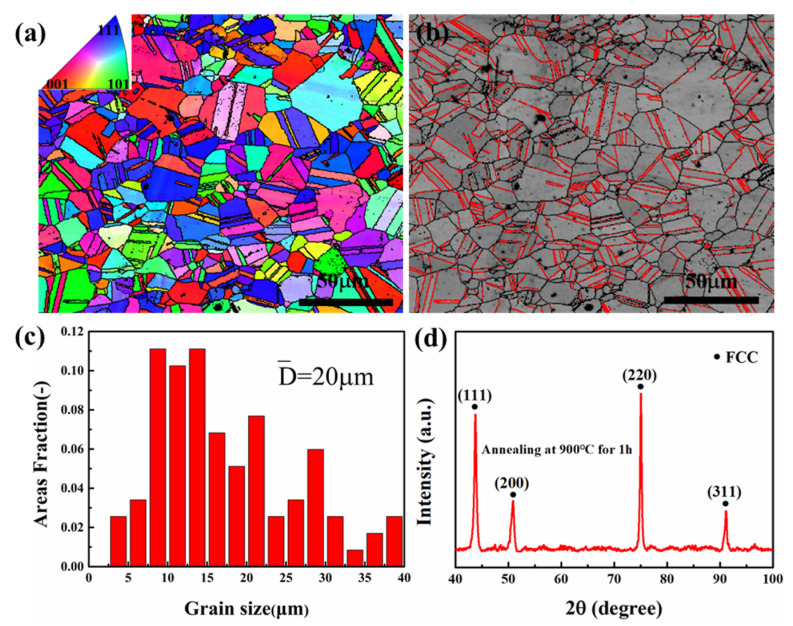
Microstructural characterization of recrystallized CoCrNiAl_0.1_Si_0.1_ MEA: (**a**) EBSD inverse pole figure map, (**b**) twins band contrast map, (**c**) grain size distribution chart, and (**d**) X-ray diffraction pattern.

**Figure 2 materials-14-07574-f002:**
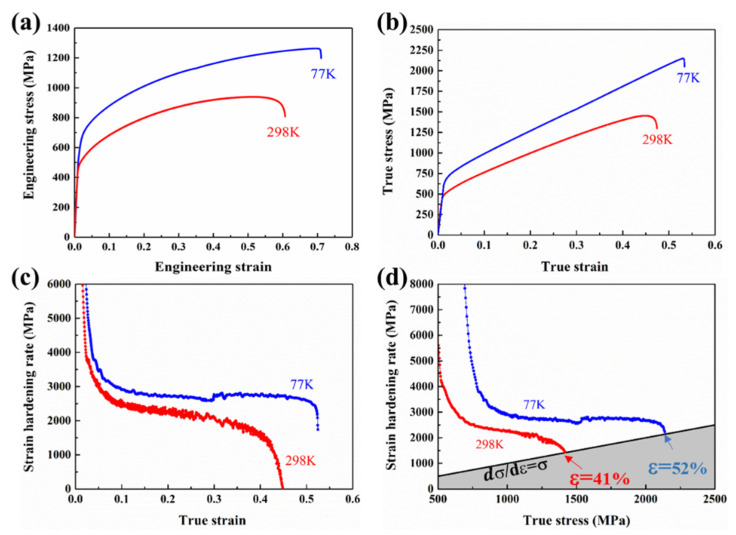
(**a**) Engineering stress–strain curves, (**b**) true stress–strain curves, (**c**) work hardening rate versus true strain curves, and (**d**) work hardening rate versus true stress curves.

**Figure 3 materials-14-07574-f003:**
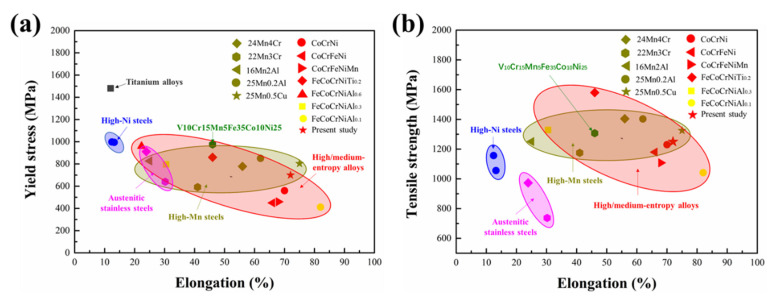
The maps of tensile yield strength versus elongation (**a**) and tensile strength versus elongation (**b**) at 77 and 110 K for HEAs in comparison to conventional cryogenic alloys [6,8,10,11,24,25].

**Figure 4 materials-14-07574-f004:**
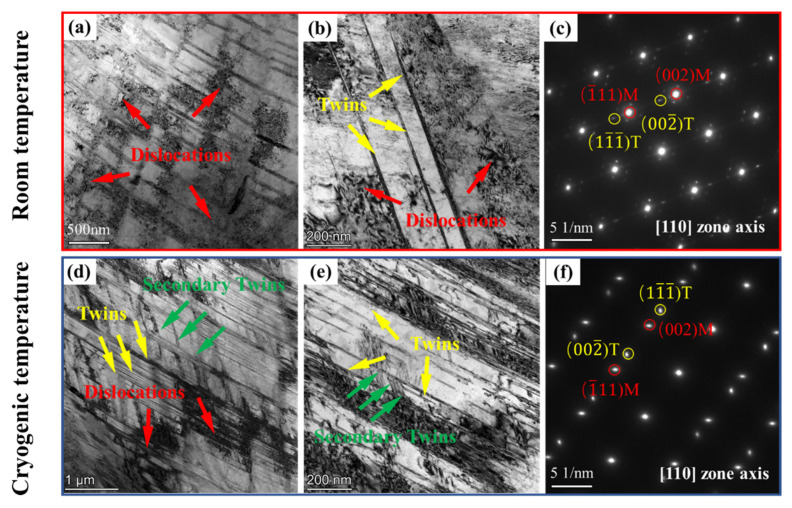
TEM images of the deformation microstructures upon (**a**–**c**) room temperature (298 K) and (**d**–**f**) cryogenic temperature (77 K). (**a**,**b**,**d**,**e**) Bright field (BF) images; (**c**,**f**) the corresponding selected area electron diffraction (SAED) pattern.

**Figure 5 materials-14-07574-f005:**
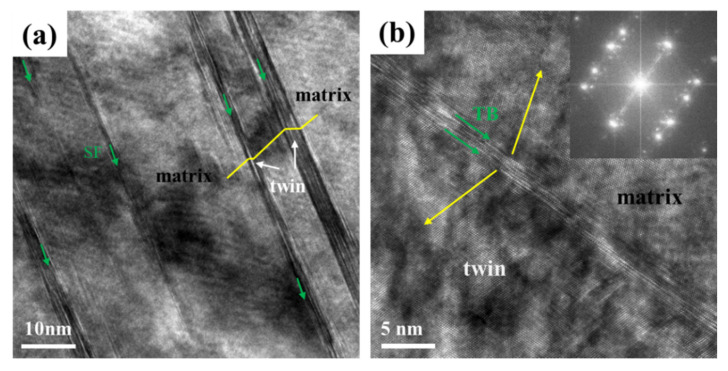
HRTEM images of the CoCrNiAl_0.1_Si_0.1_ for the experiments conducted at cryogenic temperature: (**a**) nanotwinning, TBs, and SFs; (**b**) twins, TBs, and IFFT.

**Figure 6 materials-14-07574-f006:**
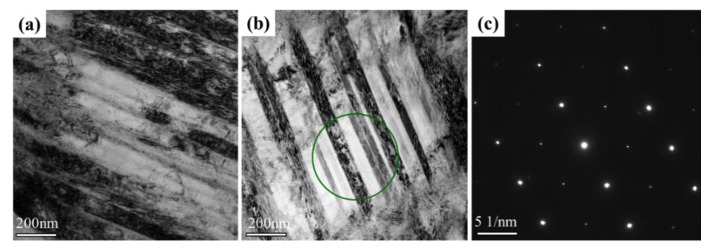
TEM image of microbands: (**a**) bright-field image showing the extent of microband, (**b**) bright-field image reconstructed showing another microband region, (**c**) corresponding <110> SEAD showing no twinning reflections.

**Figure 7 materials-14-07574-f007:**
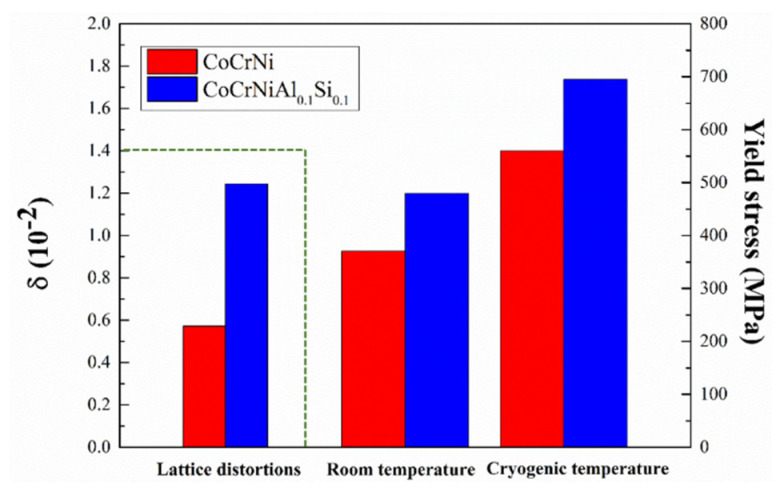
Comparison of lattice distortions, and room and cryogenic temperature yield stress for CoCrNi and CoCrNiAl_0.1_Si_0.1_.

**Figure 8 materials-14-07574-f008:**
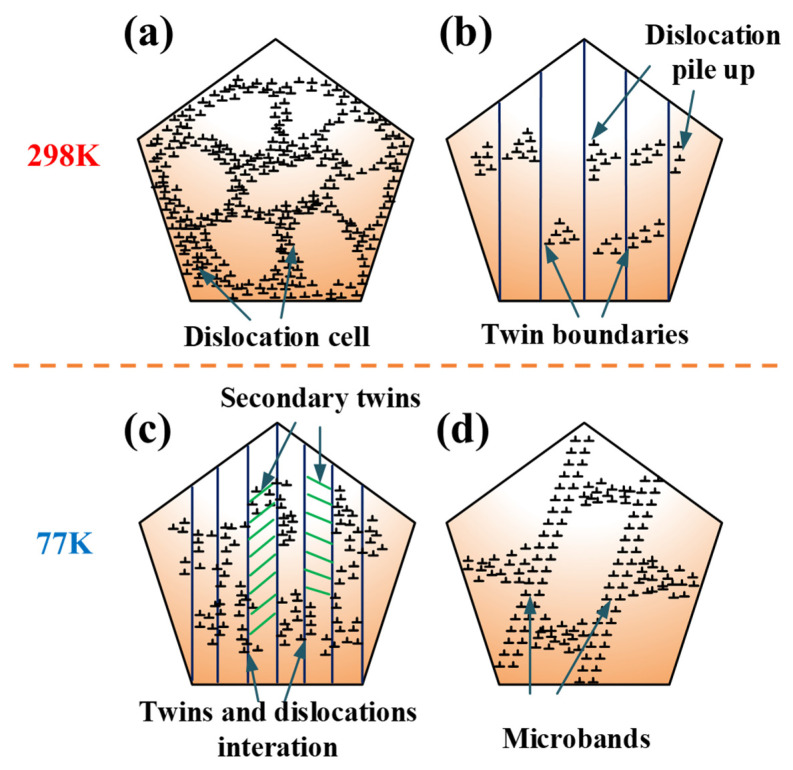
Schematic diagram of deformation mechanisms at 29 K and 77 K: (**a**) dislocation cells; (**b**) dislocations pile-up against twin boundaries; (**c**) interaction of twins, secondary twins, and dislocations; (**d**) microbands.

**Table 1 materials-14-07574-t001:** EDS chemical composition analysis (atom %).

Alloy	Co	Cr	Ni	Al	Si
CoCrNiAl_0.1_Si_0.1_	32.13%	31.29%	30.54%	3.11%	2.93%

**Table 2 materials-14-07574-t002:** Yield strength (σ_y_), ultimate tensile strength (σ_UTS_), and elongation (ε) obtained at room temperature (298 K) and cryogenic temperature (77 K).

Alloys	σ_y_(MPa)	σ_UTS_(MPa)	ε
298 K	480	900	58%
77 K	700	1250	72%

**Table 3 materials-14-07574-t003:** Thermodynamic numerical values and functions used for the calculations.

Paramaters	Thermodynamic Function (J/mol)	References
${\Delta G}_{Co}^{\gamma \to \varepsilon }$	−427.5 + 0.615T	[8]
${\Delta G}_{Cr}^{\gamma \to \varepsilon }$	1370 − 0.613T	[8]
${\Delta G}_{Ni}^{\gamma \to \varepsilon }$	1046 + 1.255T	[14]
${\Delta G}_{Al}^{\gamma \to \varepsilon }$	2800 + 5T	[14]
${\Delta G}_{Si}^{\gamma \to \varepsilon }$	−1800 + T	[14]
$\Omega_{CoCr}^{\gamma \to \varepsilon }$	2158.24	[38]
$\Omega_{CoNi}^{\gamma \to \varepsilon }$	−820 − 1.65T	[39]
$\Omega_{CrNi}^{\gamma \to \varepsilon }$	4190	[40]
σ	15	[14]

## Data Availability

Not applicable.

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
