# Peer review of "Enhanced Strength and Plasticity of CoCrNiAl0.1Si0.1 Medium Entropy Alloy via Deformation Twinning and Microband at Cryogenic Temperature"

_materials, 2021, doi:10.3390/ma14247574_

Round 1
Reviewer 1 Report
This work is very interesting and I have no remarks in the merit area. Only some small explanations must be added (eg. #15 and #18). The research programme and justification is well. Nevertheless, many editorial improvements are needed.
Generally editorial remarks:
1 - In many places Present Simple Tense is used instead Past Simple/Past Perfect. It is in places, when the research procedures are described eg. lines: 115, 188
2 - "temperature" and "microstructure" should be rather in Singular than in Plural
3 - please, check the value of "room temterature" in whole work, becauce in some places is 298 K and in others is 293 K.
4 - numarous editorial errors are in work : dots instead commas; Capital letters instead Lowercase letters and vice versa;
5 - the shortcut "SF" (line 79) is not explained. Is it stacking fault?
Detailed remarks
#1 lines 110 & 113 - the unit should be "mm" not "mm3"
#2 line 111-112 - delete "along the length" because it is confusing "thickness (...) was reduced (...) along the length"
#3 line 112 - change "annealed" and "quenched" to "annealing" and "quenching"
#4 line 118 lack of unit of strain rate (add "s-1"); temperature "298 K" - in abstract is "293 K" - check it in whole article if is correct
#5 line 120 - not "repaired" but "repeated"
#6 line 123 - lack of the space between "JEOL" and "JSM", JEOL is a company
#7 line 124 - after "Diffraction" add "detector", because EBSD is a technique, not equipment
#8 line 126 - HCLO4 correct to HClO4
#9 line 127-132 Almost the same phrase is repeated in this part of text "cut from a uniformly deformed field near the fracture surface", please to correct it
#10 line 139 - there is "dot" and should be a "comma"
#11 line 145 - wrong sign of Angstrem (not capital A, lack of small circle above) - place the valua of Angstrem also in metric unit [A=10-10 m]
#12 line 146 - change "percent" to "percentage"
#13 table 1 - the sum of all elements is 99,9%
#14 lines 154-157 - I suggest to place these data in table
#15 line 160 - dot after CoCrNiFe
#15 In why way the true strain and true stress were measured, especially in liquid nitrogen? Was used some method of calculating these values from engineering strain/stress and other mechanical properties? It must be described in article in details
#16 lines: 171, 177, 178 183 - see general remarks 3 and 4
#17 lines 173-174 - I suggest to change % to decimal values (eg. 10-1), in comparison to data at Fig. 2.
#18 Figure 3 - where are data at 77 K and 110 K ?? some explanation
#19 line 202 - correct sentence "Twins thickness and spacing are both are relatively wide." I don't understand it.
#20 lines 211-213 - see general remarks 2 and 4
#21 line 222- should be "TEM images..."
#22 line 231 - should be "MEA has" or "MEAs have" (plural/singular)
#23 line 298 - the equation is (not "are")
#24 lines 302-305 - wrong written "Hall-Petch"; what is "d" in equation; insert numbering of the equation; from where are values of σfr?
#25 line 312 - correct "ith" to "ith"
#26 line 315 - wrong Figure munber; should be 7th
#27 Figure 7 - Divide into two graphs (a) and (b), because there are two values in different units, but the colors and the way of presentation are the same. It will be more clear.
#28 lines 348, 367, 371, 374 - see general remarks 2 and 4 (line 374 "Where" change to "where", and there is dot, too)
#29 line 377 - change "a" into "a"
#30 line 381, 383 - correct "Xi" and "Thermodynamic" into "Xi" and "thermodynamic"
#31 line 389-390 - Second part of sentence has not predicate
#32 lines 400, 406, 409, 412, 414, 415 - see general remarks 2, 3, 4 (aditionaly: line 400 "diagrams", line 412 "conclusions" - plural)
#33 line 413 - correct the sentence eg. "In this study, CoCrNiAl0.1Si0.1 MEA, manufactured by a vacuum arc-melting furnace, was investigated "
#34 point 4.2 lines 422-426 - too long sentence (four lines); it is difficult to understand the meaning of the sentence
#35 lines 427-428 - the chic in the sentence is incorrect
Reviewer 2 Report
It is interesting and concise article but the following errors or doubts should be improved or explained:
- There are some minor errors in the text. All the text must be checked and improved:
- line 14: “design” instead of “desigen”,
- line 15: “annealing” instead of “anneal”,
- line 45: “plasticity enhancement” can be replaced by “plasticity increase”.
- The sum of the components (table 1) should be 100%.
- How many specimens of each material and process combination were tested in the tensile tests? How was the strain measured? Was it taken from crosshead displacement or by extensometer?
- In their conclusions, the authors should express an opinion on the intended use of the tested materials (more precisely than described in lines 46 and 47.
Reviewer 3 Report
The work reported in the manuscript is described in the last paragraph of the introduction as showing better performance of alloys containing Al and Si than pure CrCoNi and as a contribution to understanding temperature-dependent deformation mechanisms.
No experimental results are given for pure CrCoNi alloy. It is thus difficult to access the effect of the Al and Si additives. Figure 3 indicates there is little difference between the pure alloy and alloy with the additives. Comparable performance of the pure alloy at ambient temperature is confirmed in several references, such as referece 10 in their Figure 5 with engineering strain 900 MPa and stress 58%, and Y L Zhao et al., Heterogeneous precipitation behavior and stacking-fault-mediated deformation in a CoCrNi-based medium-entropy alloy, Acta Materialia 138 (2017) 72-82, in their Figure 6 with stress 900 MPa and strain 55%. It is thus questionable what the Al and Si additives are contributing to performance.
Line 267 states that the pure alloy exhibits significant strengthening as the temperature is lowered. The role of Al and Si again comes into question since line 418 describes the alloy with additives as exhibiting remarkable enhancement of strength and ductility at low temperatures. Reference 10 reports 1200 Mpa and 75% at 77K in Figure 13 for the pure alloy, in line with Figure 2a in the manuscript.
The contributing factors to the deformation mechanisms described in point 3 of the conclusion have been reported previously. For example Z Zhang et al., Dislocation mechanisms and 3D twin architectures generate exceptional strength-ductility-toughness combination in CrCoNi medium-entropy alloy, Nature Communications 8 (2016) 14390 which describes twin networks having pathways for dislocation glide and cross slip between twin boundaries observed at ambient temperature by in situ TEM. Reference 10 takes up the same increase in deformation twinning and dislocation slip at cryogenic temperatures as in point 3.
In its present form the manuscript does not clearly show what the additives are contributing to alloy performance or what new information is being reported with respect to deformation mechanisms.
Round 2
Reviewer 3 Report
There is still some question if the Si and Al additives are necessary to improve performance of the CrCoNi base alloy. The Zhao 2017 reference describes preparing CoCrNi "by arc-melting a mixture of high-purity raw elements (>99.9% pure) in a Ti-gettered argon atmosphere. The ingot was flipped and remelted five times to promote chemical homogeneity" which is similar to what is stated in section 2. Their Fig 6a shows performance for the base alloy on a par with what is presented for the Si and Al additives.
Reference 38 is mentioned on lines lines 59, 366, 372 and Table 3 but nothing is said about it containing control data for CoCrNi and CoCrNiSi0.1. Figs 1 and 2 in the authors' response show performance for the base CrCoNi and with Si additive but are not included in the manuscript despite the vague statement of adding "this part of the data in the paper". It should at least be mentioned somewheres that control experiments are presented in the reference.
The authors' response emphasizes that microbands have not been reported previously for CoCrNi alloys. This is not entirely true since Z Wu et al., Enhanced strength and ductility of a tungsten-doped CoCrNi medium-entropy alloy, Journal of Materials Research (2018) , report observing "a few microbands" and not much twinning at room temperature. The manuscript reports in a similar unconvincing fashion on line 255 that "besides, microbands were found" and it will be left to future studies to confirm the observations.
